# Nomogram for Predicting Survival in Patients Treated with Liposomal Irinotecan Plus Fluorouracil and Leucovorin in Metastatic Pancreatic Cancer

**DOI:** 10.3390/cancers11081068

**Published:** 2019-07-28

**Authors:** Li-Tzong Chen, Teresa Macarulla, Jean-Frédéric Blanc, Beloo Mirakhur, Floris A. de Jong, Bruce Belanger, Tanios Bekaii-Saab, Jens T. Siveke

**Affiliations:** 1National Institute of Cancer Research, National Health Research Institutes, Tainan 704, Taiwan; 2Division of Hematology/Oncology, Department of Internal Medicine, National Cheng Kung University Hospital, Tainan 704, Taiwan; 3Vall d’Hebron University Hospital and Vall d’Hebron Institute of Oncology, 08035 Barcelona, Spain; 4Groupe Hospitalier Haut-Lévêque, CHU Bordeaux, 33600 Pessac, France; 5Ipsen Biopharmaceuticals, Inc., Basking Ridge, NJ 07920, USA; 6Global Medical Affiars, Servier, 8002 Zürich, Switzerland; 7Ipsen Biopharmaceuticals, Inc., Cambridge, MA 02142, USA; 8Mayo Clinic Cancer Center, Phoenix, AZ 85054, USA; 9German Cancer Consortium (DKTK) and German Cancer Research Center (DKFZ), 69120 Heidelberg, Germany; 10West German Cancer Center, University Hospital Essen, 45147 Essen, Germany

**Keywords:** liposomal irinotecan, NAPOLI-1, nomogram, pancreatic cancer, survival outcomes

## Abstract

NAPOLI-1 (NCT01494506) was a phase III study of liposomal irinotecan (nal-IRI) plus 5-fluorouracil/leucovorin (5-FU/LV) in patients with metastatic pancreatic ductal adenocarcinoma (mPDAC) previously treated with gemcitabine-based therapy. This post hoc analysis of NAPOLI-1 aimed to develop a predictive nomogram for overall survival (OS) at 6 and 12 months. Analyses were derived from all patients in NAPOLI-1 randomized to receive nal-IRI+5-FU/LV, nal-IRI monotherapy, or 5-FU/LV combination therapy. OS was associated with baseline factors using univariate and multivariable Cox analyses. A predictive nomogram was derived and validated using a concordance index and calibration plots. The univariate analyses identified 21 independent factors that contributed to OS, with eight factors significantly associated with OS. The Karnofsky Performance Score contributed the largest number of points (100), followed by presence of liver metastasis (98) and randomization to nal-IRI+5-FU/LV (96). The other baseline factors showing effects were albumin (g/dL), neutrophil/lymphocyte ratio, carbohydrate antigen 19-9 (U/mL), disease stage at diagnosis, and body mass index (kg/m^2^). The nomogram was used to predict the 6- and 12-month survival probability. The mean absolute errors between the observed and predicted probabilities for OS at 3, 6, and 9 months were 0.07, 0.08, and 0.07, respectively. This nomogram, based on NAPOLI-1, provides additional insight to aid decision-making for patients with mPDAC after previous gemcitabine-based therapy.

## 1. Introduction

Recently released data for the global cancer burden in 2018 show that pancreatic cancer is the seventh most frequent cause of cancer-related mortality worldwide [1]. Regional mortality rates due to pancreatic cancer are higher in the United States (third) and Europe (fourth), with Asia (seventh) reporting similar rates to global estimates [2,3,4]. Although there has been a steady decline in cancer-related deaths worldwide for many other types of cancer, deaths from pancreatic cancer have remained unchanged over the past 10 years [3,5]. The 5-year overall survival (OS) rate is poor, at approximately 5–8% [5,6]. Surgical resection at early diagnosis is the only potentially curative therapy; however, 70–80% of patients are diagnosed at a later stage with unresectable disease and more than 50% of patients present with metastatic disease.

Two recommended front-line regimens for fit patients with locally advanced or metastatic pancreatic ductal adenocarcinoma (mPDAC) include gemcitabine plus nab-paclitaxel or fluorouracil-based combination regimens with FOLFIRINOX (a combination of oxaliplatin, folinic acid, irinotecan, and fluorouracil) [7,8]. The choice of a second-line regimen for patients with mPDAC who experience disease progression usually includes chemotherapy agents that were not included in the first-line regimen. The available treatment options for a second-line regimen depend on several factors, such as the patient’s Eastern Cooperative Oncology Group Performance Status (ECOG PS), comorbidities, organ function, patient tolerance for cytotoxic regimens, and patient preference and support system.

A nomogram is a graphical representation of a statistical predictive model that generates a numerical probability of a clinical event [9]. Nomograms provide an opportunity to evaluate multiple patient-based factors and to predict a patient’s individualized risk for a prespecified event, such as survival, death, or a particular adverse event of concern [10,11]. Nomograms have become increasingly important in oncology for providing more decision-making tools for the individualized treatment of advanced cancer [12,13,14,15].

Liposomal irinotecan (nal-IRI) consists of irinotecan, a topoisomerase I inhibitor, encapsulated in liposomes [16], which was designed to utilize the enhanced permeability and retention effect of nal-IRI in tumors. The retention of nal-IRI is thought to increase availability of irinotecan’s active metabolite, SN-38, in the targeted tumor [16,17,18,19,20]. Pharmacokinetic analyses have demonstrated prolonged circulation of the irinotecan-containing liposome compared with free irinotecan, which results in prolonged exposure of SN-38 in the tumor [16,20,21,22]. The clinical impact of the irinotecan-containing liposome was demonstrated in NAPOLI-1 (NCT01494506), a phase III, double-blind, randomized study where patients with mPDAC who were previously treated with gemcitabine-based therapy achieved a significant improvement in OS (primary endpoint), progression-free survival (PFS), and objective response rate with nal-IRI+5-fluorouracil/leucovorin (5-FU/LV) versus 5-FU/LV alone [23]. Furthermore, treatment with nal-IRI+5-FU/LV resulted in a manageable safety and tolerability profile in this trial.

Adherence to National Comprehensive Cancer Network (NCCN) Guidelines^®^ for treating pancreatic adenocarcinoma has improved survival; however, developing nomograms to identify key factors affecting OS may improve individualized management of patients with pancreatic cancer [7,10]. The objective of this exploratory post hoc analysis of the NAPOLI-1 trial is to develop a nomogram that attempts to predict OS in patients with metastatic adenocarcinoma of the pancreas after disease progression following gemcitabine-based therapy. The factors analyzed were based on the baseline patient characteristics and other variables from the trial. The specific aim was to develop a nomogram that predicts OS at 6 and 12 months for patients treated with nal-IRI+5-FU/LV based on the NAPOLI-1 trial data [23].

## 2. Materials and Methods

### 2.1. Patients

The NAPOLI-1 study methods (Protocol No./IRS No.: MM-398-07-03/B-BR-100-160; date: 5 April 2012) have been described previously [23]. In brief, this study comprised patients aged 18 years of age or older with confirmed pancreatic adenocarcinoma and documented distant metastatic disease and disease progression after previous gemcitabine-based therapy. This post hoc analysis included all patients in the trial who were randomized to receive either nal-IRI+5-FU/LV combination therapy, nal-IRI monotherapy, or 5-FU/LV combination therapy. The post hoc analyses were based on the primary survival analysis data for the NAPOLI-1 clinical trial, which was conducted from 11 January 2012 through 14 February 2014.

### 2.2. Statistical Analyses

A statistical analysis plan was developed, based on the use of univariable and multivariable analyses, to determine the factors within the NAPOLI-1 clinical trial that were significantly predictive of OS shown in Figure 1. A multivariable Cox model on OS was developed using baseline factors that were significantly predictive of OS by univariable analysis (*p* < 0.10) or that were considered to be clinically important and retained in the model after stepwise selection (*p* < 0.10). The final set of identified factors was used to create a nomogram that assigned points equal to the weighted sum of the relative significance of each factor. The most predictive factor was assigned a maximum point value of 100, and points for other factors were determined based on a comparison with the most influential factor.

Validation of the final model was assessed by concordance index (c-index), which was bias-corrected using a bootstrap (2000 iterations) methodology. The total patient population was divided into low-, middle-, and high-risk categories by allocating the total points from all patients into tertiles and survival by risk group and displayed via Kaplan‒Meier plots.

## 3. Results

### 3.1. Patient Distribution and Demographics

The patients (*N* = 417) who entered the NAPOLI-1 trial were randomly assigned to nal-IRI+5-FU/LV (*n* = 117), nal-IRI monotherapy (*n* = 151), or FU/LV (*n* = 149). The baseline patient characteristics in NAPOLI-1, reported previously, were balanced among treatment arms [23].

### 3.2. Prognostic Factors of Overall Survival

The predictive value of factors affecting OS were based on data from the univariate analysis (*n* = 417) and multivariate analysis (*n* = 399) of patients, with *n* = 18 patients excluded because of missing baseline data. Univariate analyses were used to evaluate 21 factors for association with OS (Table 1). The eight clinically relevant variables that were found to be significantly associated with OS (or within a proximity to the prespecified α level) in the univariate analysis were used in the multivariate analysis (all *p* < 0.01, except body mass index (*p* = 0.03)). Based on the findings of the original multivariable analysis, the multivariate Cox regression was repeated to ensure the clinical relevance of the variables (Table 2). A second multivariate Cox regression was generated, excluding treatment, which modified the model slightly but did not impact the overall findings.

### 3.3. Nomogram for OS

The multivariate analysis showed that the Karnofsky Performance Score (KPS) contributed the largest number of points (100) to the predicted OS, followed by presence of liver metastasis (98) and randomized treatment arm to nal-IRI+5-FU/LV (96). The other five factors that showed effects were baseline albumin (g/dL), neutrophil/lymphocyte ratio, baseline cancer antigen 19-9 (CA19-9, U/mL), disease stage at diagnosis, and body mass index (kg/m^2^) (Figure 2).

### 3.4. Utilizing the Nomogram for OS

Each clinically relevant variable in the nomogram was evaluated for each patient and assigned a score. The total points score for individual patients was determined by summing their scores for all variables. The total points score corresponds to the predicted OS probability at 6 and 12 months. A larger total points sum on the nomogram corresponds to a greater 6- and 12-month survival probability.

### 3.5. Model Validation

The total patient population was divided into tertiles to create risk categories as follows: low risk (>370 points, *n* = 131), intermediate risk (260–370 points, *n* = 137), and high risk (<260 points, *n* = 131). OS outcome was differentiated among the risk tertiles. Figure 3 shows the Kaplan‒Meier plots for each risk group; the median OS values of the low-, medium-, and high-risk groups were 8.5, 5.3, and 2.9 months, respectively. The mean absolute errors between the observed and predicted probabilities for OS at 3, 6, and 9 months were 0.07, 0.08, and 0.07, respectively. The estimate of the bias-corrected c-index for the final nomogram for OS was 0.70 (95% confidence interval (CI): 0.67–0.73).

To evaluate the effect of nal-IRI+5-FU/LV treatment on survival outcomes within each individual risk group, a separate prognostic nomogram was developed to predict 6- and 12-month OS excluding the treatment variate (Figure 4). Kaplan‒Meier curves for OS of patients from each treatment arm were developed for each risk group (Appendix A). In the high-risk group, patients in the 5-FU/LV, nal-IRI monotherapy, and nal-IRI+5-FU/LV treatment arms had median OS values of 2.5, 2.8, and 4.3 months, respectively. Patients with medium risk had median OS values of 4.3, 4.9, and 5.4 months for 5-FU/LV, nal-IRI monotherapy, and nal-IRI+5-FU/LV treatment, respectively. Patients with low risk in the 5-FU/LV, nal-IRI monotherapy, and nal-IRI+5-FU/LV treatment arms had median OS values of 6.4, 7.2, and 9.0 months, respectively.

## 4. Discussion

Although advances in therapies for pancreatic cancer have improved OS and other outcomes, patients with pancreatic cancer are often diagnosed late in an advanced stage and have a poor prognosis compared with other types of cancer [7]. The Tumor Node Metastasis (TNM) staging system has limitations with respect to accounting for other contributing factors that affect OS in patients with pancreatic cancer, since other determinative factors—such as performance status—are not taken into account with TNM staging [11]. Using a nomogram to identify patient risk for OS may provide a better basis for informed treatment decisions, including the decision to treat with nal-IRI+5-FU/LV.

Previous nomograms have been developed for evaluating OS in pancreatic cancer [24,25,26,27]. Vienot and colleagues developed a nomogram to predict OS in patients with advanced pancreatic ductal adenocarcinoma before receiving second-line therapy. Their nomogram was designed to identify low-, intermediate-, and high-risk groups [24]. Another nomogram was developed by Goldstein and colleagues based on predicting OS in patients treated with nab-paclitaxel plus gemcitabine or gemcitabine alone for metastatic pancreatic cancer. In this analysis, the factors predicting OS were the treatment arm, KPS, neutrophil-to-lymphocyte ratio, albumin level, sum of longest tumor diameters, and presence of liver metastasis [25]. Other studies have used multivariate analysis to determine that a better KPS and lower CA19-9 were related to OS in locally advanced pancreatic cancer and in those with metastatic pancreatic cancer. In addition, younger age, better performance status, peritoneal metastasis, higher serum albumin, lower CA19-9, and CA19-9 variation were found to be related to OS [28].

The NAPOLI-1 clinical trial stepwise Cox regression analysis of nal-IRI+5-FU/LV versus 5-FU/LV identified baseline characteristic factors associated with OS, such as KPS, that were included as important factors for the nomogram developed in this study. There was some variability compared with the current analysis in the sets of factors examined, with the current multivariable analyses not designed to generate individualized patient prediction of OS. The eight factors that were determined to be the most influential on OS from the NAPOLI-1 study were KPS, presence of liver metastasis, randomized treatment arm to nal-IRI+5-FU/LV, baseline albumin (g/dL), neutrophil/lymphocyte ratio, baseline CA19-9 (U/mL), disease stage at diagnosis, and body mass index (kg/m^2^), which are similar to the factors found in other nomograms for predicting OS in metastatic pancreatic cancer [24,25]. These factors are also aligned with baseline and prognostic characteristics identified in the COnsensus statement on Mandatory Measurements in unresectable PAncreatic Cancer Trials (COMM-PACT) as mandatory for future pancreatic cancer trials [29].

The eight clinically relevant variables identified here were used to develop a prognostic nomogram that aimed to predict OS at 6 and 12 months based on risk classification, derived from baseline factors and treatment received. A nomogram that excluded treatment options as a variable was developed to assess the impact of second-line chemotherapy treatment options on OS for patients in each risk group. In the model with treatment options removed, the nomogram indicated that treatment with nal-IRI+5-FU/LV resulted in consistently greater OS across risk groups compared with treatment with nal-IRI or 5-FU/LV alone.

A limitation of this nomogram analysis is the lack of an external validation group. However, the large patient population in the NAPOLI-1 clinical trial assisted in overall validation in this analysis. The c-index for the final nomogram for OS was 0.70 (95% CI: 0.67–0.73), which indicates good discrimination ability and was also in the range of other nomograms. C-indices of 0.69 (95% CI: 0.67–0.71) and 0.75 (95% CI: 0.72–0.78) were obtained for the Goldstein and Vienot studies, respectively [24,25]. Another limitation was the uneven distribution of the range of some factors for patients who entered the NAPOLI-1 study. There were fewer patients with KPS <70, albumin <30 g/L, or increased bilirubin in the NAPOLI-1 trial [23].

## 5. Conclusions

Univariable and multivariable analyses identified eight baseline patient and disease characteristics from patients in the NAPOLI-1 clinical trial that formed the basis of a nomogram to assist in predicting the OS of mPDAC patients who had progressed on prior gemcitabine-based treatments. Treatment with nal-IRI+5-FU/LV was identified as an important factor in the nomogram. The use of this nomogram to distinguish between risk groups may aid in clinical decision-making for this patient population.

## Figures and Tables

**Figure 1 cancers-11-01068-f001:**
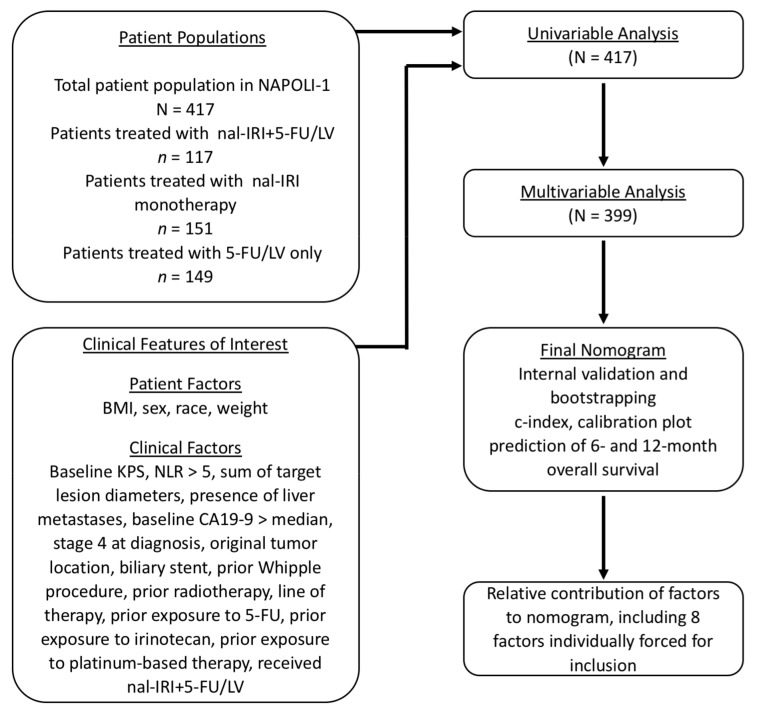
Statistical analysis plan.

**Figure 2 cancers-11-01068-f002:**
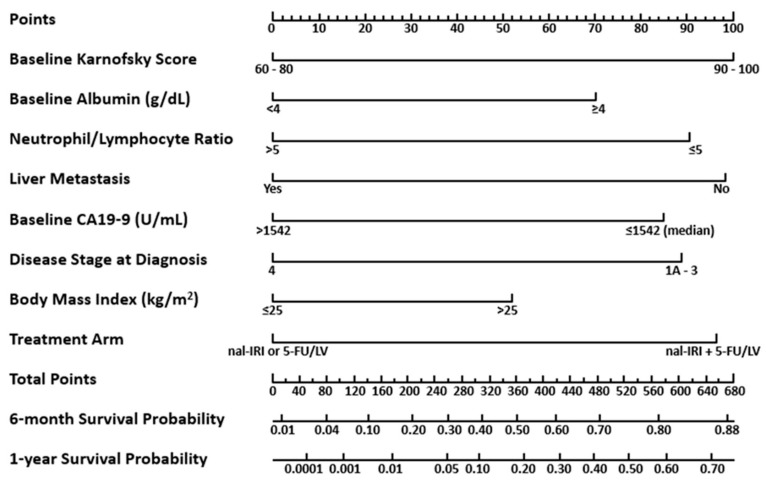
Nomogram to predict overall survival in patients with metastatic pancreatic cancer treated with nal-IRI+5-FU/LV in the NAPOLI-1 study. 5-FU/LV, 5-fluorouracil/leucovorin; CA19-9, cancer antigen 19-9; nal-IRI, liposomal irinotecan.

**Figure 3 cancers-11-01068-f003:**
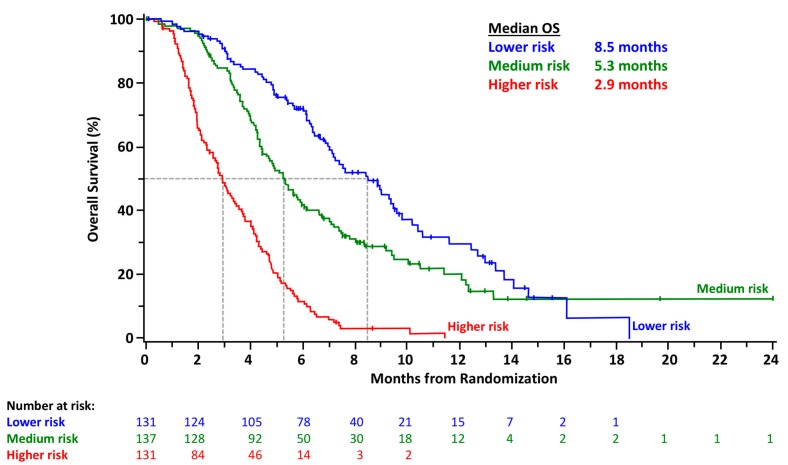
Kaplan‒Meier survival curves showing overall survival probability stratified by risk group.

**Figure 4 cancers-11-01068-f004:**
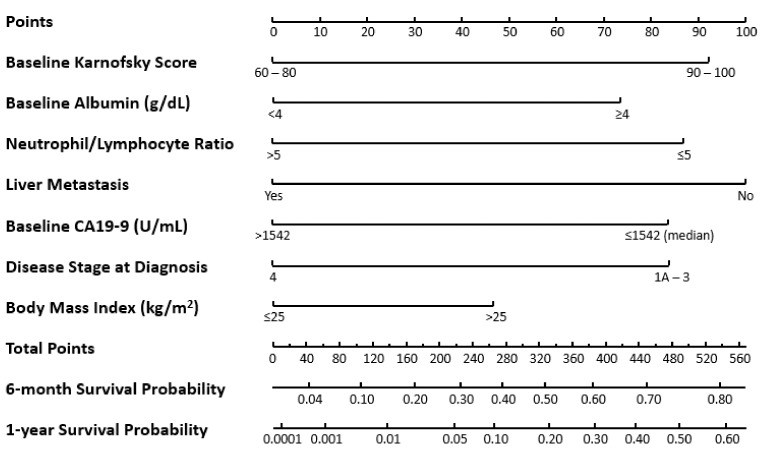
Nomogram to predict overall survival in patients with metastatic pancreatic cancer in the NAPOLI-1 study excluding treatment arm. CA19-9, cancer antigen 19-9.

**Table 1 cancers-11-01068-t001:** Univariate Cox regression of overall survival (*N* = 417).

Parameter, M (SD)	*n*	Wald *p*-Value	Hazard Ratio (95% CI)
Baseline Karnofsky Performance Score ^1^, ≥90 vs. <90	232, 185	<0.0001	0.527 (0.421, 0.660)
Baseline albumin, ≥4 g/dL vs. <4 g/dL	227, 190	<0.0001	0.643 (0.515, 0.802)
Neutrophil/lymphocyte ratio, ≤5 vs. >5	292, 123	<0.0001	0.582 (0.458, 0.741)
Sum of longest diameter of target lesions (mm), 71.4 (46.31)	417	<0.0001	1.005 (1.003, 1.007)
Presence of liver metastases, yes vs. no	285, 132	<0.0001	1.688 (1.314, 2.168)
Baseline CA19-9, >median (1542 U/mL) vs. ≤median	202, 202	<0.0001	1.620 (1.291, 2.032)
Stage 4 disease at time of diagnosis ^2^, yes vs. no	213, 200	<0.0001	1.774 (1.413, 2.226)
Primary tumor location: Head of pancreas vs. other	256, 161	0.19	0.860 (0.685, 1.079)
Prior biliary stent, yes vs. no	37, 380	0.90	0.973 (0.651, 1.455)
Prior Whipple procedure, yes vs. no	113, 304	0.021	0.739 (0.573, 0.955)
Prior radiotherapy, yes vs. no	97, 320	0.0046	0.668 (0.506, 0.883)
Prior line of therapy in metastatic setting, (0, 1, 2+)	51, 234, 132	0.71	1.033 (0.870, 1.227)
Prior exposure to 5-FU, yes vs. no	183, 234	0.43	1.095 (0.874, 1.370)
Prior exposure to irinotecan, yes vs. no	46, 371	0.13	1.324 (0.922, 1.902)
Prior exposure to platinum-based therapy, yes vs. no	137, 280	0.44	1.099 (0.867, 1.394)
Received nal-IRI+5-FU/LV, yes vs. no	117, 300	0.0008	0.640 (0.493, 0.830)
Age (years), 62.8 (9.68)	417	0.13	1.009 (0.997, 1.020)
Body mass index >25 kg/m^2^ vs. ≤25 kg/m^2^	122, 295	0.020	0.746 (0.584, 0.954)
Race: White vs. non-white	253, 164	0.30	1.129 (0.900, 1.416)
Race: Asian vs. non-Asian	136, 281	0.20	0.857 (0.677, 1.085)
Sex: Female vs. male	180, 237	0.86	0.979 (0.782, 1.227)
Weight (kg), 65.3 (15.66)	417	0.065	0.993 (0.987, 1.000)

^1^ Where a baseline Karnofsky Performance Score was missing, assignment was based on category per randomization. ^2^ In NAPOLI-1, all patients had metastatic disease (stage 4) at time of study entry. 5-FU/LV, 5-fluorouracil/leucovorin; CA19-9, cancer antigen 19-9; CI, confidence interval; M, mean; nal-IRI, liposomal irinotecan; SD, standard deviation.

**Table 2 cancers-11-01068-t002:** Multivariate Cox regression of overall survival (*N* = 399) ^1^.

Parameter	Eight-Parameter Model				Seven-Parameter Model		
	Number of Patients	Parameter Estimate (β)	Wald *p*-Value	Hazard Ratio (95% CI)	Parameter Estimate (β)	Wald *p*-Value	Hazard Ratio (95% CI)
Baseline Karnofsky score ≥90 vs. 60–80	219/180	–0.545	<0.0001	0.58 (0.46, 0.74)	–0.502	<0.0001	0.61 (0.478, 0.768)
Baseline albumin ≥4 vs. <4 g/dL	221/178	–0.382	0.0013	0.68 (0.54, 0.86)	–0.398	0.0008	0.67 (0.532, 0.848)
Neutrophil/lymph- ocyte ratio ≤5 vs. >5	284/115	–0.493	0.0001	0.61 (0.47, 0.79)	–0.471	0.0002	0.63 (0.486, 0.802)
No liver metastasis vs. liver metastases	124/275	–0.534	<0.0001	0.59 (0.45, 0.76)	–0.541	<0.0001	0.58 (0.449, 0.754)
Baseline CA19-9 ≤1542 vs. >1542 U/mL	199/200	–0.462	<0.0001	0.63 (0.50, 0.79)	–0.454	0.0001	0.63 (0.504, 0.801)
Stage <4 vs. Stage 4 at diagnosis ^2^	190/209	–0.483	<0.0001	0.62 (0.49, 0.78)	–0.454	0.0002	0.63 (0.501, 0.805)
Body mass index >25 vs. ≤25 kg/m^2^	121/278	–0.283	0.030	0.75 (0.58, 0.97)	–0.252	0.052	0.78 (0.603, 1.002)
nal-IRI+5-FU/LV vs. 5-FU/LV or nal-IRI	112/287	–0.523	0.0001	0.59 (0.45, 0.77)			

^1^ Model excludes 18 patients with a missing value of one or more parameters. ^2^ In NAPOLI-1, all patients had metastatic (stage 4) disease at time of study entry. 5-FU/LV, 5-fluorouracil/leucovorin; CA19-9, cancer antigen 19-9; CI, confidence interval; nal-IRI, liposomal irinotecan.

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
