# Peer review of "Nomogram for Predicting Survival in Patients Treated with Liposomal Irinotecan Plus Fluorouracil and Leucovorin in Metastatic Pancreatic Cancer"

_cancers, 2019, doi:10.3390/cancers11081068_

Reviewer 1 Report

Overall, I think this is an interesting and important paper that would be suitable for Cancers.

Focused features, statistic models and the nomograph which authors proposed were appropriate.

Authors addressed the limitations of the analysis, but the process of analysis and results are valuable for readers of Cancers.

Author Response

The authors would like to thank the reviewer for their positive feedback.

Reviewer 2 Report

The authors present a well designed nomogram for survival in pancreatic cancer, incorporating the potential benefits of chemotherapy, on the basis of retrospective analysis of data from an RCT. this is a clinically useful tool to translate trial results and statistics into practice.

Author Response

(The authors gave the same response as above.)

Reviewer 3 Report

This study is a sub-study of the NAPOLI-1 trial describing the use of liposomal irinotecan in stage 4 pancreatic cancer patients. The authors have analyzed prognostic factors and integrated these in a nomogram to predict survival. 417 patients from the three study arms were analyzed and 399 with complete data integrated in the analysis. The authors show that Karnofsky Performance Score, presence of liver metastasis, randomization to nal-IRI+5-FU/LV, albumin, neutrophil/lymphocyte ratio, CA 19-9, disease stage at diagnosis and body mass index are the variables that predict survival accurately. After exclusion of the type of chemotherapy, the remaining 7 factors can serve as pre-therapeutic prognostic predictors and 3 specific subgroups can be defined according to the nomogram.

The manuscript is interesting in terms of a pre-therapeutic approach to stratify patients.

Points to address:

In a prior analysis (Wang-Gillam et al., Eur J Cancer 2019), the same group has shown an age-cutoff of 65 years as a prognostic factor. Why was this not included in the present work?

A basic question – also in the prior publications of the NAPOLI trial – is the definition of “metastatic”. In the present work, the authors define 190/399 patients as<stage 4 (table 2). From my understanding <stage 4 is not metastatic, i.e. regional lymph nodes are not regarded as a metastatic stage. Referring to the publication by Wang-Gillam on the same patients, this needs to be explained and commented. In this latter publication, only 52 patients showed regional lymph node metastases, presumably qualifying these patients as <stage 4. In the present work, 190 patients are defined as <stage 4? Were intrapancreatic metastases defined as <stage 4? If so, this is questionable from my point of view as well – these are rather synchronous cancers than metastases.

I would suggest including the supplemental 7-factor nomogram in the main manuscript instead of the 8-factor nomogram. The pre-therapeutic factors excluding the chemotherapy regimen are more important to give this nomogram a clinical value.

Author Response

Comment 1: In a prior analysis (Wang-Gillam et al., Eur J Cancer 2019), the same group has shown an age-cutoff of 65 years as a prognostic factor. Why was this not included in the present work?

Response: In the analysis of Prognostic Factors of Overall Survival (see Section 3.2), the predictive value of age was included as part of the univariate Cox regression. In this analysis, age (evaluated as a continuous variable, not with a cutoff) was not found to be a predictive factor (P=0.13), and so was not included in the multivariate analysis.

Action: Table 1 was updated with the n per subgroup for each of the binary parameters evaluated; where continuous variables were evaluated, the mean and standard deviation was added to the parameter column.

Comment 2: A basic question – also in the prior publications of the NAPOLI trial – is the definition of “metastatic”. In the present work, the authors define 190/399 patients as Stage 4 (table 2). From my understanding Stage 4 is not metastatic, i.e. regional lymph nodes are not regarded as a metastatic stage. Referring to the publication by Wang-Gillam on the same patients, this needs to be explained and commented.

Response: In NAPOLI-1 study, patients may have had<stage 4 (non-metastatic) diseases at the time of diagnosis (before their front-line therapy) but all the patients had to have metastatic (stage 4) diseases at study entry (when they failed to respond to previous treatment, including gemcitabine-based treatment). Therefore, “stage at diagnosis” versus “metastatic stage (at study entry)” may need to be further emphasized. In addition, regional lymph node metastases could occur in patients with metastatic pancreatic cancer but it is usually considered a component of locally advanced disease by itself.   

In order to have been included in the NAPOLI-1 study, patients were required to have:

1.     Histologically or cytologically confirmed adenocarcinoma of exocrine pancreas

2.     Documented metastatic disease; disease status was permitted to be measurable or non-measurable as defined by RECIST v. 1.1 guidelines

3.     Documented disease progression after prior gemcitabine or gemcitabine-containing therapy, in locally advanced or metastatic setting.

Per the 7th Edition of the American Joint Committee on Cancer on Pancreatic Cancer Stage (which were utilized as the benchmark for staging), stage 4 disease indicates that the cancer has spread to distant sites such as the liver, peritoneum (the lining of the abdominal cavity), lungs or bones (M1). It can be any size (Any T) and might or might not have spread to nearby lymph nodes (Any N). https://www.cancer.org/cancer/pancreatic-cancer/detection-diagnosis-staging/staging.html

As per the inclusion criteria, the inclusion of patients with regional lymph node involvement was dependent upon meeting all three criteria and not solely on the absence or presence or regional lymph nodes.

Action: A footnote has been added to Tables 1 and 2 to clarify that stage 4 was required for NAPOLI-1 study entry.

Comment 3: In this latter publication, only 52 patients showed regional lymph node metastases, presumably qualifying these patients as Stage 4. In the present work, 190 patients are defined as << span="">Stage 4?

Response: In NAPOLI-1 study, patients may have had<stage 4 (non-metastatic) diseases at the time of diagnosis (before their front-line therapy) but all the patients had to have metastatic (stage 4) disease at study entry (when they failed to respond to previous treatment, including gemcitabine-based treatment). As per the inclusion criteria, the inclusion of patients with regional lymph node involvement was dependent upon meeting all three criteria described above and not solely on the absence or presence or regional lymph nodes.

Action:No additional text recommended

Comment 4: Were intrapancreatic metastases defined as Stage 4? If so, this is questionable from my point of view as well – these are rather synchronous cancers than metastases.

Response: Clinically, synchronous or intra-pancreatic metastatic pancreatic cancers are extremely rare and difficult to diagnose. Patients in the NAPOLI-1 study had pancreatic lesion(s), but none were reported as intra-pancreatic metastasis. Although pancreas was listed as a “site of metastatic disease” in the previous NAPOLI-1 publications (Lancet 2016, EJC 2019), rather than “intra-pancreatic metastasis,” they may have been more appropriately labelled as lesions within the pancreas, either initially unresectable or local recurrence/relapse after previous definitive local therapy.  

Action: No additional text recommended

Comment 5: Comment: I would suggest including the supplemental 7-factor nomogram in the main manuscript instead of the 8-factor nomogram. The pre-therapeutic factors excluding the chemotherapy regimen are more important to give this nomogram a clinical value.

Response: The authors agree to include the 7-factor nomogram in the main manuscript, as suggested by the reviewer.

Action: 7-factor nomogram incorporated into the main manuscript as Figure 4.